# Data-driven aeolian dust emission scheme for climate modelling, evaluated with EMAC 2.55.2

Klaus Klingmüller[1] and Jos Lelieveld[1,2]

[1]Max Planck Institute for Chemistry, Hahn-Meitner-Weg 1, 55128 Mainz, Germany
[2]The Cyprus Institute, P.O. Box 27456, 1645 Nicosia, Cyprus

**Correspondence:** Klaus Klingmüller (k.klingmueller@mpic.de)

**Abstract.** Aeolian dust has significant impacts on climate, public health, infrastructure and ecosystems. Assessing dust concentrations and the impacts is challenging because the emissions depend on many environmental factors and can vary greatly with meteorological conditions. We present a data-driven aeolian dust scheme that combines machine learning components and physical equations to predict atmospheric dust concentrations and quantify the sources. The numerical scheme was trained to reproduce dust aerosol optical depth retrievals by the Infrared Atmospheric Sounding Interferometer on board the MetOp-A satellite. The input parameters included meteorological variables from the fifth generation atmospheric reanalysis of the European Centre for Medium-Range Weather Forecasts. The trained dust scheme can be applied as an emission submodel, to be used in climate and Earth system models, which is reproducibly derived from observational data so that a priori assumptions and manual parameter tuning can be largely avoided. We compared the trained emission submodel to a state-of-the-art emission parametrisation, showing that it substantially improves the representation of aeolian dust in the global atmospheric chemistry-climate model EMAC.

## 1 Introduction

Aeolian dust is one of the most abundant aerosol components worldwide and substantially affects the Earth system in many ways. In contrast to sea salt, the only other component contributing a comparable fraction to the total aerosol loading, aeolian dust is emitted over land, where accordingly the highest dust loads occur and severely interfere with human health and activities. The generally high and often exceptional particulate matter concentrations in the vicinity of dust sources cause both, acute health problems (Goudie, 2014) and contribute to the long-term exposure with associated health risks and excess mortality (Lelieveld et al., 2019b, a). Reduced visibility during dust events can interrupt road and air traffic, cause accidents and reduce solar electricity production, the latter aggravated by deposited dust, resulting in high economical costs (Middleton et al., 2019). Since the particles interact with radiation and clouds, atmospheric dust significantly affects weather and climate. Deposited on snow and ice, mineral dust reduces the surface albedo and accelerates glacier melting (Di Mauro et al., 2019; Francis et al., 2022). On the other hand, deposited dust particles are an important source of mineral nutrients, both on land and in the ocean (Bristow et al., 2010). Although aeolian dust predominantly originates from natural sources, they can be affected by

anthropogenic climate change and land use (Klingmüller et al., 2016), moreover atmospheric dust interacts with anthropogenic air pollution (Klingmüller et al., 2019, 2020). Consequently, aeolian dust is not a purely natural phenomenon.

These are only some of the diverse aspects which constitute the great relevance of aeolian dust in Earth science and drive the demand for accurate representations in atmospheric models. However, meeting this demand is challenging as can be seen, for example, from the large uncertainty range of model estimates for the global annual dust emission (Huneeus et al., 2011). Aeolian dust emissions depend on many environmental factors. Considering that some of those have been identified only recently (Rodriguez Caballero et al., 2022), presumably not all relevant factors and their impacts are known or well quantified. Temporally changing environmental factors have to be considered, which can vary within seconds (wind and turbulence), seasons (vegetation), years (land cover) and geological time scales (soil composition). Quantitative observations of dust emissions are difficult and thus a direct comparison of modelled emission fluxes with observations, especially on global scale, not feasible. Therefore, dust emission models can only be validated indirectly based on atmospheric dust concentration observations or scarce dust deposition measurements, which requires that atmospheric processing and transport are taken into account.

We present a novel approach to address these challenges combining two developments which have become main drivers of Earth system science, the increasing availability of satellite observations and the advancing machine learning technology (Reichstein et al., 2019; Bauer et al., 2021; Eyring et al., 2021; Irrgang et al., 2021). We used the resulting, hybrid dust emission scheme to substantially improve the representation of aeolian dust in the global atmospheric chemistry-climate model EMAC (Jöckel et al., 2006), demonstrating a valuable synergy of machine learning and physical process based modelling.

This article is structured as follows: The datasets are described in section 2. The architecture of the data-driven dust model is presented in section 3, its training and evaluation in sections 4 and 5. Section 6 documents the EMAC setup used in section 7 to demonstrated the benefits of the newly derived dust emissions. Conclusions are drawn in section 8.

## 2 Data

Our selection of input variables follows that of common online dust emissions schemes (e.g., Astitha et al., 2012; Klingmüller et al., 2018). Data for several input variables were taken from the fifth generation European Centre for Medium-Range Weather Forecasts (ECMWF) atmospheric reanalysis (ERA5), including the single-level variables surface friction velocity, total precipitation, snow depth, volumetric soil water in the topmost soil layer, leaf area index (LAI) of low vegetation, LAI of high vegetation and surface geopotential (Hersbach et al., 2018b), and the height dependent variables of the u-component of wind, v-component of wind and vertical velocity (Hersbach et al., 2018a). The latter variables where considered at four pressure levels, 650 hPa, 750 hPa, 850 hPa and 950 hPa, to represent the atmosphere up to about 4 km altitude. Mineral dust can reach higher altitudes, but since most of the dust mass remains within these layers, this approximation is a reasonable compromise between a realistic representation of dust transport and an acceptable computational burden. For all time dependent variables, hourly data were considered. Horizontally, we re-gridded the data to a regular longitude-latitude grid at 1° resolution.

The ERA5 data were complemented by the static soil clay fraction distribution from the SOILPOP30 data base (Nickovic, 2011) and the annual Moderate-resolution Imaging Spectroradiometer (MODIS) collection 6 International Geosphere–Biosphere Programme (IGBP) land surface class fractions (Friedl and Sulla-Menashe, 2015), both re-gridded to the same 1° grid.

As output variable representing the atmospheric dust distribution we consider the dust aerosol optical depth (DAOD) which is closely related to the vertically integrated amount of dust and can be retrieved remotely from satellites so that datasets with

consistent global coverage are available. We used version 8 of the 10 $\mu$m DAOD product developed at the Université libre de Bruxelles (ULB) based on observations by the Infrared Atmospheric Sounding Interferometer (IASI) on board the MetOp-A satellite (Clarisse et al., 2019). In the infrared, coarse particles scatter radiation more efficiently than submicron particles. Therefore, observations at a wavelength of 10 $\mu$m are most sensitive to mineral dust and sea salt and are little affected by other aerosols, resulting in a particularly reliable DAOD retrieval. We considered all years which are completely covered by

the dataset, in total 12 years from 2008 to 2019. Years with incomplete coverage and the continuation of the product based on retrievals from the MetOp-B satellite were omitted. The level 2 data were split into hourly data by rounding the observation times to the nearest full hour and gridded to the 1° grid of the input data by averaging all retrievals covered by each grid cell. Grid cell values below the 0.05 % percentile and above the 99.95 % percentile were omitted.

Towards the poles, less DAOD retrievals are obtained, the grid cell geometry increasingly deviates from a rectangular shape

that we assume in our dust scheme, and the Courant–Friedrichs–Lewy criterion gains importance, while only a small fraction of the global dust is transported to or emitted from these regions. Therefore we restricted the domain of all data sets to latitudes between 60°S and 60°N.

We assume the aerosol extinction contributing to the DAOD to be proportional to the dust concentration. Accordingly, the total amount of dust over each grid cell is proportional to the DAOD and the horizontal area of the grid cell. By multiplication

with the latitude dependent cell area, we converted the DAOD to a quantity proportional to the total amount of dust over each grid cell, which, after normalisation, corresponds to the vertical integral of the dust representation in our model.

Surface friction velocity, soil moisture, snow depth, LAI, geopotential and clay fraction were normalised by subtracting their mean and division by their standard deviation. Dust amount and precipitation were only divided by their standard deviation to preserve their sign. This normalisation maps all input variables and associated gradients into a similar and relatively small

range to support the model parameter optimisation routine during the training process.

The 12 years of DAOD data and the corresponding input data were split into a validation period (2008), a test period (2009) and a training period (2010 to 2019). While we used the validation period to monitor the training procedure and validate the model development, the test period was exclusively used to evaluate the final, trained model. Only the training period was used for model training. The order of the periods was chosen such that reliable emission data for EMAC was readily available

for the test period, but at the same time the training period could be easily extended in the course of the model development. Moreover, the validation period before the test period serves as spin-up period of the EMAC simulations.

## 3 The data-driven dust model

The data-driven dust model combines multiple modules representing different processes, including dust emission, transport and removal. Figure 1 provides an outline of the model architecture. The processes are applied iteratively once every hour, synchronous with the hourly time step of the input and output data. Consistent with the input wind fields, the instantaneous dust concentration is implemented as 3 dimensional array $(d_{ijk})$, where $i = 1, \cdots, n_{\text{lon}}$, $j = 1, \cdots, n_{\text{lat}}$ and $k = 1, \cdots, n_{\text{lev}}$, corresponding to a grid with 1° horizontal spacing ($n_{\text{lon}} = 360$, $n_{\text{lat}} = 120$) and $n_{\text{lev}} = 4$ vertical layers. All $n_{\text{lon}} \times n_{\text{lat}} \times n_{\text{lev}}$ array elements $d_{ijk}$ are initialised with 0. We used $n_{\text{t}} = 168$ iterations, corresponding to a time interval of 168 h, i.e., 7 days.

Centrepiece of the model is the emission module which itself consists of three components. At each horizontal grid cell $(i, j)$, an interim emission flux $s_{ij}$ is calculated by a stack of densely connected neural layers using the 6 input variables surface friction velocity, LAI of low vegetation, LAI of high vegetation, soil moisture, snow cover and clay fraction. This generic deep neural network architecture is computationally efficient but still very powerful for various classification and, as in our application, regression problems.

The stack consists of 2 hidden layers of 64 units each with leaky rectified linear unit (ReLU) activations, $\text{LeakyReLU}(x) = \max(0, x) + 0.01 \min(0, x)$, and a final output unit with softplus activation, $\text{Softplus}(x) = \ln(1 + \exp(x))$, to ensure positive output. The non-zero gradient of the leaky ReLU function at negative values avoids vanishing gradients that can hinder the model optimisation process when using the ordinary ReLU activation function, which is zero at negative values. Within the layers, each unit calculates the weighted sum of its input variables, where the weights are trainable parameters, adds a trainable bias, and applies the activation function before passing the result to the next layer. In the densely connected stack, all units in one layer are connected to all units in the next layer. The emission module is translation invariant in both space and time because identical trainable weights and biases are used for all locations $(i, j)$ and time steps. Therefore, spatial and temporal variations of the modelled emissions are based solely on changing input parameters.

The effect of the topography is considered by multiplying the interim emission flux with a topography dependent factor $\beta_{\text{topo},ij}$, similarly to the use of topographic source functions in existing dust emission schemes (Ginoux et al., 2001; Klingmüller et al., 2018),

$$s_{ij} \leftarrow \beta_{\text{topo},ij} s_{ij}. \tag{1}$$

Unlike the intermediate emission flux, which depends only on the local input parameter values, the topography effect considers the surrounding terrain to account for, e.g., the accumulation of sediments in valleys and depressions. In deep learning, such dependency on surrounding values is commonly addressed by convolutional layers which consider all values from grid cells defined by the convolution kernel and apply trainable weights and biases which are translation invariant. The topography factor $\beta_{\text{topo},ij}$ is computed from the geopotential height field by a stack of convolutional layers with 3 by 3 grid cell kernels. The stack consists of 3 hidden layers with 4 output channels each and leaky ReLU activations, and a 1 channel output layer with sigmoid activation, $\text{sig}(x) = (1 + \exp(-x))^{-1}$, to limit the output to the interval (0, 1). This architecture computes the topography factor for each grid cell based on the geopotential in the surrounding 9 by 9 grid cells, i.e., a 9° by 9° region, similarly to the 10° by 10° region used by Ginoux et al. (2001).

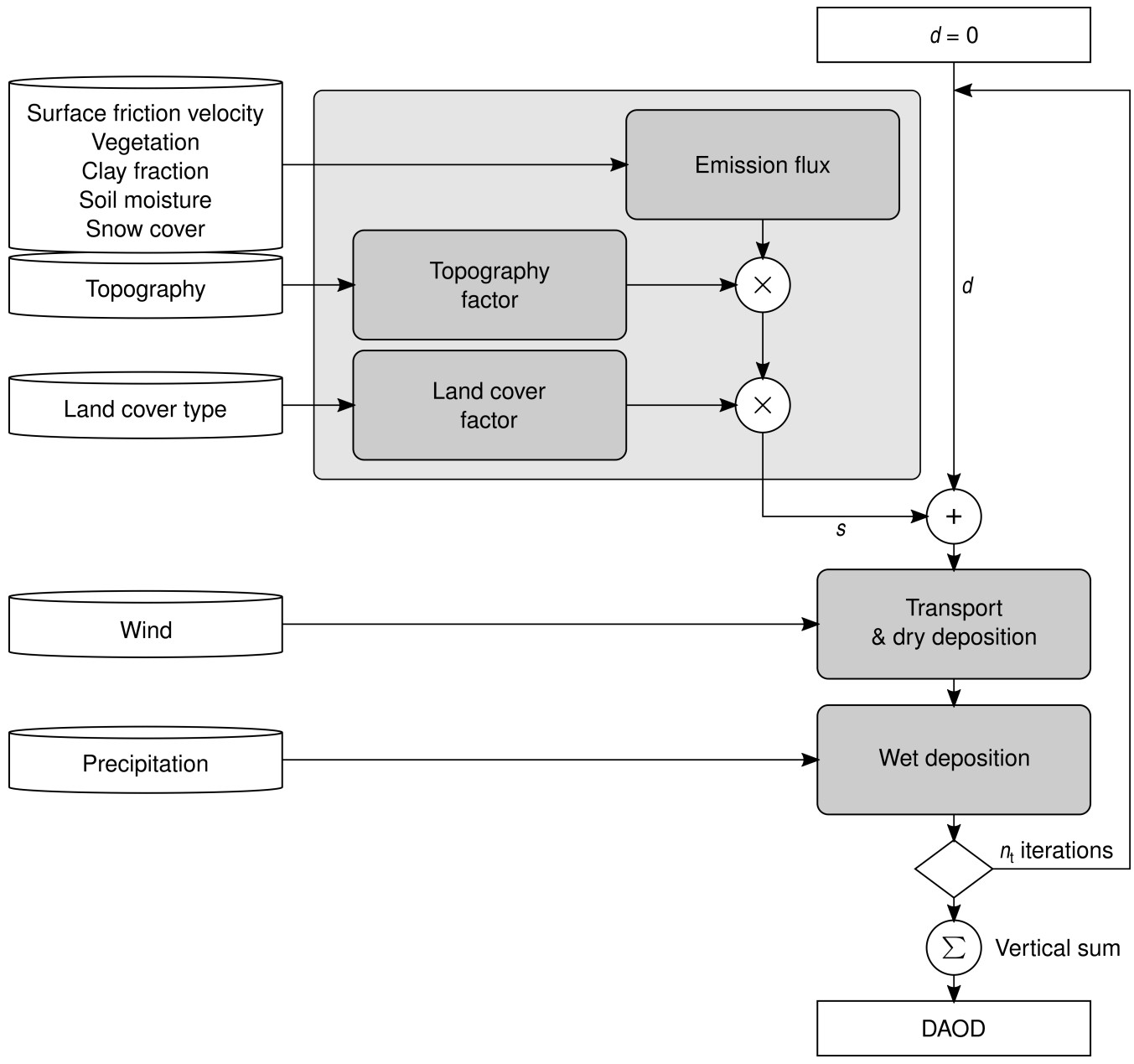

**Figure 1.** Outline of the dust model. It combines multiple modules representing different processes and effects (small grey boxes), each with trainable parameters. The dust emissions are represented by three submodels (large grey box).

The size of the convolutional network and the densely connected neural network, i.e. the number of layers and number of units or channels per layer, were adjusted based on experiments with different settings. The model should be large enough to provide sufficient capacity to learn the relevant relationships, but not larger than necessary to avoid overfitting and reduce computational requirements. Model performance is not very sensitive to limited changes in model size, but could be improved by systematic optimisation of the latter.

Also the land cover effect is considered by multiplying the interim emission flux with a corresponding factor, generalising the concept of the land cover class based emission mask used by Klingmüller et al. (2018), for example. The land cover factor is represented by a single linear layer with 16 inputs representing the 16 IGBP land cover classes. The layer multiplies the fraction $f_{ijl}$ of grid cell $(i,j)$ that is covered by class $l$ with a trainable, class specific emission efficiency $\beta_{\mathrm{lc},l}$ ranging from 0 to 1 and sums over all classes to obtain the average emission efficiency of the grid cell $\beta_{\mathrm{lc},ij} = \sum_{l=1}^{16} f_{ijl}\beta_{\mathrm{lc},l}$. The interim emission flux is multiplied by this emission efficiency to obtain the final emission flux,

$$s_{ij} \leftarrow \beta_{\mathrm{lc},ij} s_{ij}. \tag{2}$$

According to this emission flux, the dust concentration in the lowest vertical layer ($k = n_{\mathrm{lev}}$) of the dust array is increased,

$$d_{ijn_{\mathrm{lev}}} \leftarrow d_{ijn_{\mathrm{lev}}} + s_{ij}A_{ij}, \tag{3}$$

where $A_{ij}$ is the grid cell area, before the dust array is processed further by the transport module.

The transport module successively applies vertical diffusion, vertical wind transport, north-south transport and east-west transport. The vertical diffusion considers that some fraction of the dust is exchanged vertically between grid cells independent of the large scale vertical velocity of the air masses, e.g., by convection or sedimentation, and is parametrised by two trainable parameters, the total fraction of dust leaving the grid cell, and the fraction thereof moving upwards. The transport by wind in each of the 3 spatial dimensions is calculated by considering the overlap of the grid cells translated according to the wind speed and the time step length of 1 hour with the neighbouring cells, and has no trainable parameters. The wind speed components are limited so that the Courant number $C \leq 1$. In the vertical transport calculation, dry deposition is considered by allowing a fraction of the dust transported onto the surface to remain on the surface, the rest is retained in the lowest dust layer. The fraction represented by the latter is defined by a trainable parameter. However, during training this parameter typically quickly converges to 1, indicating that the current implementation of our approach is not sensitive enough to identify dry deposition. The reason is likely the limited atmospheric residence time discussed below, representing an unrealistic additional removal process. Moreover, a fraction of the deposition that always occurs collocated with emissions might be represented by a reduced emission flux.

As dominant removal process we consider wet deposition. Every hourly time step, the dust concentration in all vertical layers is reduced by a fraction proportional to the local total precipitation $p_{ij}$,

$$d_{ijk} \leftarrow (1 - \beta_{\mathrm{p}} p_{ij}) d_{ijk}, \tag{4}$$

where the proportionality factor $\beta_{\mathrm{p}}$ is a trainable parameter. This approximation considers the correlation between precipitation and dust removal as the dominant statistical relationship with the strongest influence on dust source parameters during

training. More detailed parametrisations could improve future versions of the model. In addition, the finite number of iterations considered by the model implies a corresponding maximum atmospheric residence time for the dust. Here, the use of $n_t = 168$ iterations corresponds to a maximum residence time of 1 week. In reality, dust particles can reside much longer in the atmosphere, but because the contribution of those particles to the global DAOD distribution is comparatively small, most DAOD features and the dominant dust sources can be identified regardless of this restriction. However, a noticeable bias of the DAOD predictions is expected in regions remote from dust sources.

The 3 dimensional dust field is summed along the vertical dimension to obtain the total amount of dust in the vertical column. Finally, denormalisation and division by the cell area yields the DAOD prediction.

Since by design the dust emission flux is positive and neither the transport nor the removal processes can alter the sign of the dust concentration in each grid cell, the amount of dust and the DAOD predictions are guaranteed to be positive, which is desired for a realistic model. On the other hand, the noise range of the IASI retrievals includes negative values which crucially contribute to a realistic overall DAOD level. Even though these retrievals cannot be reproduced by the model, they are included in the training process to avoid a positive bias in the DAOD predictions. In combination with the expected negative bias in remote regions, this a priori limits the agreement of predicted and observed DAOD. However, for our main purpose, the extraction of enhanced dust emissions for climate simulations, it is sufficient, as demonstrated in the following.

We implemented the model using the PyTorch machine learning framework (Paszke et al., 2019). This framework is very flexible and allows the seamless combination of the different modules described above, including standard deep learning models such as densely connected neural layers and convolutional networks, classical regression models and iterative physical processing as used in the transport module, into a single hybrid model. The autograd engine, one of PyTorch's core features, automatically differentiates this complex model with respect to the trainable parameters, and the resulting gradients are used by the optimisation routines during model training.

## 4 Training

During training, the predicted amount of dust in each vertical column is compared to the observed DAOD (the latter normalised and weighted by the grid cell area as described above) by means of a loss function.

In addition to the discrepancy between predictions and observations, the training procedure should also consider reasonable a priori assumptions: The surface friction velocity initiates and sustains the horizontal flux of saltating particles and is therefore the driving force of mineral dust emissions. Consequently, the dust emission flux generally increases with increasing surface friction velocity. In contrast, vegetation, soil moisture and snow cover reduce soil erodibility and saltation and thus the emission flux. To incorporate this information into the emission module we imposed a knowledge-based regularisation by penalising violations of these rules similar to Gupta et al. (2019). For this purpose, before each optimisation step during training, we computed the partial derivatives of the emission module with respect to the corresponding input variables for $n_s = 10^6$ randomly sampled input parameters. The values consistent with the rules were set to zero, i.e., positive derivatives with respect to the surface friction velocity and negative derivatives with respect to the high and low vegetation LAI, soil moisture and snow

depth. The mean of the absolute value of the thus adjusted derivatives, divided by the mean of the $n_{\mathrm{s}} = 10^6$ emission values, was added to the mean squared error (MSE) loss function $f_{\mathrm{MSE}}$ to obtain the total loss function,

$$f_{\mathrm{loss}} = f_{\mathrm{MSE}} + \left( \frac{1}{5n_{\mathrm{s}}} \sum_{l=1}^{n_{\mathrm{s}}} \sum_{m=1}^{5} \mathrm{ReLU}(\sigma_m \frac{\partial s_l}{\partial x_m}) \right) \left( \frac{1}{n_{\mathrm{s}}} \sum_{l=1}^{n_{\mathrm{s}}} s_l \right)^{-1}, \tag{5}$$

where $(x_m)$ are the input variables and $\sigma_m = -1$ if $x_m$ is the surface friction velocity, otherwise $\sigma_m = 1$. The total loss was then minimised using the Adam optimiser (Kingma and Ba, 2014).

To generalise the model and prevent overfitting during training, we regularised the stack of densely connected layers in the emission module by applying a 5 % dropout rate to the hidden layers (Hinton et al., 2012). The combination of machine learning components and numerical implementations of physical processes such as atmospheric transport in a single model adds a novel aspect to the overfitting problem. For pure machine learning models, analysing the validation data set is sufficient to detect overfitting. In contrast, when combined with numerics, the machine learning algorithm can reduce the validation loss further by adapting to the numerical approximations (here, e.g., maximum transport period and discretisation), which is then not necessarily based on a more realistic representation of the underlying process. To avoid this type of overfitting, we trained the model using the whole training data set only once and did not repeat the training for multiple epochs. Tests using more epochs showed that the training loss can indeed be reduced further, i.e., the DAOD predictions become more realistic, however the resulting emissions did not clearly improve the EMAC simulations.

## 5   Trained model evaluation

Figure 2 shows the hourly grid-cell DAOD values predicted by the trained model during the test period (year 2009) vs. the corresponding IASI retrievals. In the range typically observed over the dust belt between West Africa and East Asia (10 $\mu$m DAOD $\gtrsim 0.05$), the predictions are realistic. Towards smaller values, the model seems to underestimate the observations. This is partly attributed to the expected underestimation in remote areas due to the aforementioned limitation of the transport time. The large scatter of the observed values around DAOD = 0 corresponds to a maximum of the density distribution at about 0.01 on the logarithmic scale. Accordingly, observations in this range are consistent with 0 and the low model predictions. The overall good performance of the model is reflected by a high Pearson correlation coefficient of 0.502, calculated using all 14371211 prediction-observation pairs for the test data set.

The temporal correlation coefficients of the observed and predicted hourly DAOD values within each grid cell $(i, j)$, $r_{ij} = r(\mathrm{DAOD}_{\mathrm{obs},ij}, \mathrm{DAOD}_{\mathrm{pred},ij})$, are typically greater than 0.5 over the regions affected by desert dust (Fig. 3). In regions remote from dust sources, where mineral dust is of minor relevance, the low, randomly varying dust concentrations result in correlation coefficients close to zero, e.g., over large parts of the Pacific and Southern Oceans. The negative correlation coefficients over some regions, notably west of Australia over the southern Indian Ocean, result from the transport time limit in our model. They occur when enhanced DAOD values are caused by dust reaching the region after more than 1 week of transport over non-emitting regions including the ocean, while the model predicts clean air with low DAOD values. Also the daily correlation coefficient, $r(t) = r(\mathrm{DAOD}_{\mathrm{obs}}(t), \mathrm{DAOD}_{\mathrm{pred}}(t))$, quantifying the similarity of the spatial pattern of predic-

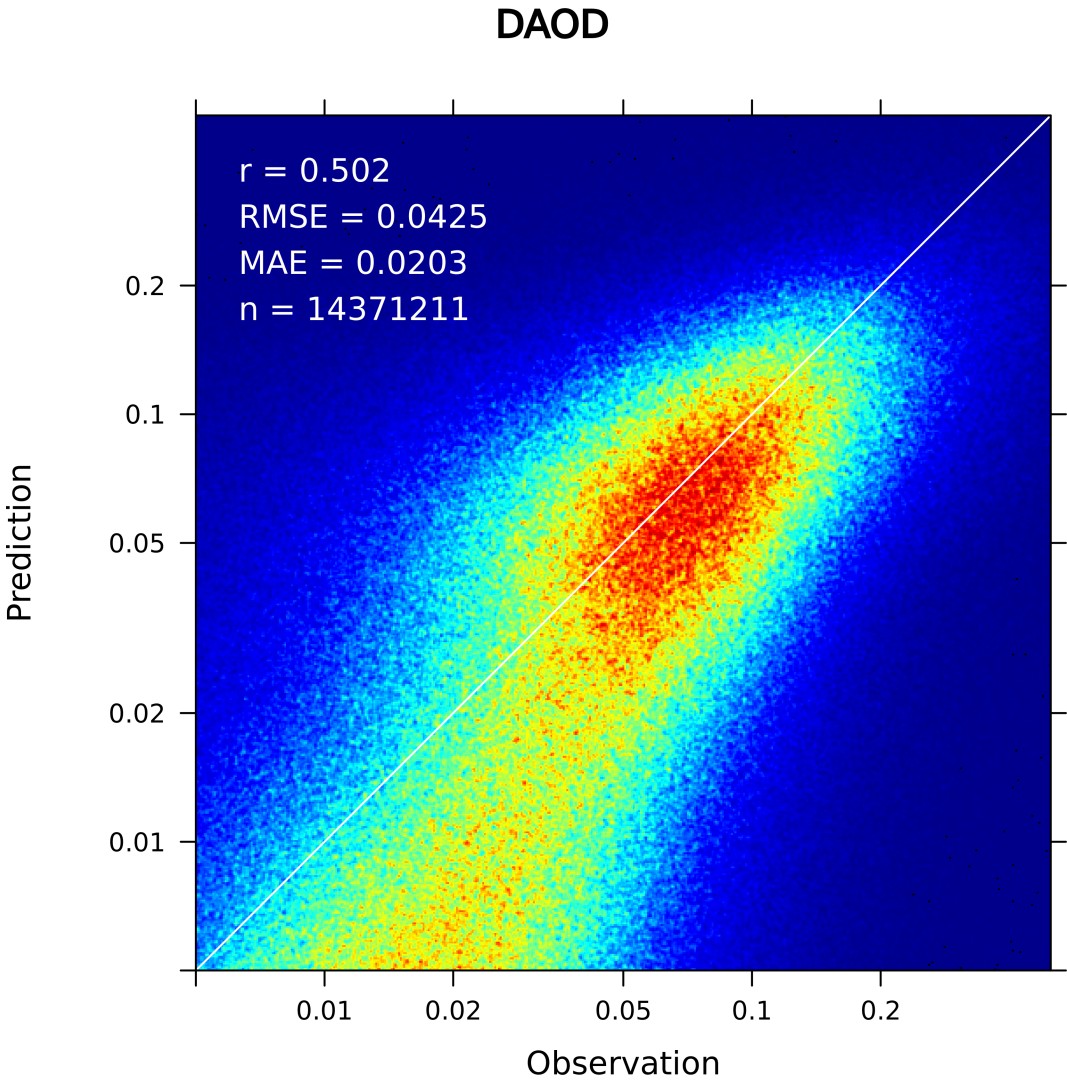

**Figure 2.** Predicted vs. observed hourly grid-cell DAOD values during the test period. The colours represent the density of data points. Correlation coefficient $r$, root-mean-square error RMSE, mean absolute error MAE and the number of data pairs $n$ are indicated.

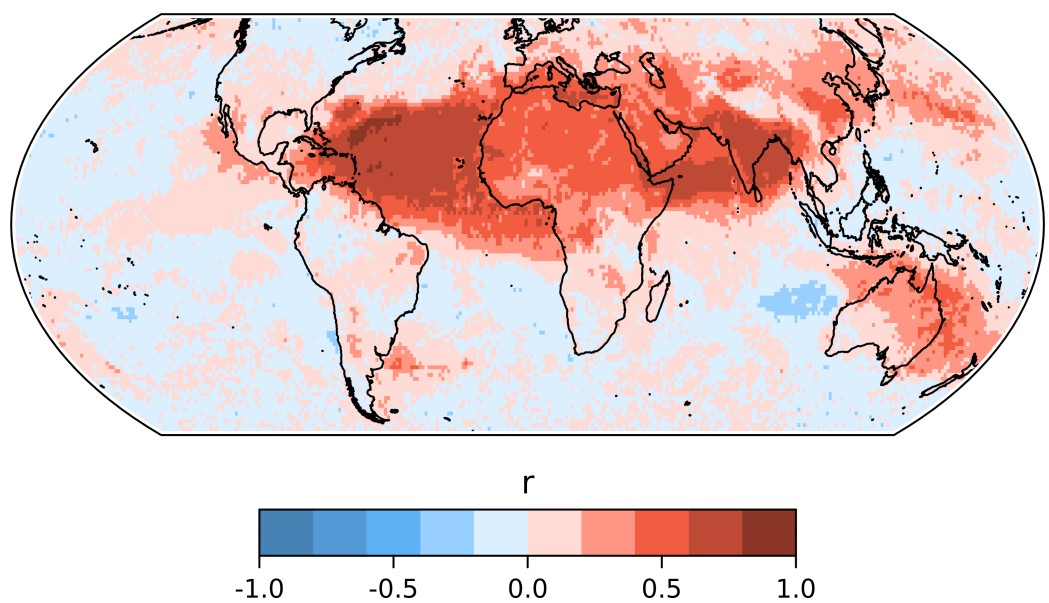

**Figure 3.** Correlation coefficient of the observed and predicted DAOD time series for each grid cell during the test period.

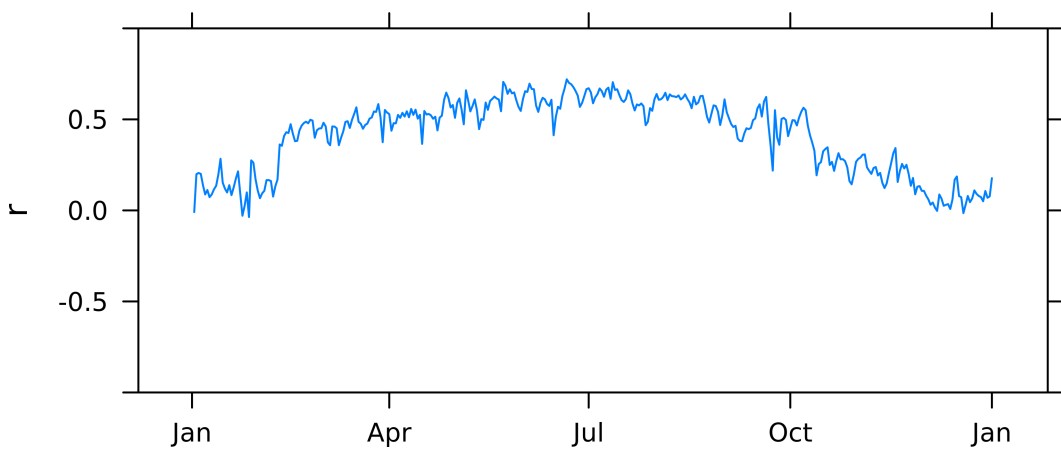

**Figure 4.** Daily correlation coefficient of the observed and predicted DAOD during the test period.

tions and observations on day $t$, reaches or exceeds 0.5 during the dust season in northern hemispheric spring and summer
(Fig. 4). During winter, the decline of dust activity and thus the absence of distinct spatial patterns results in lower correlation coefficients.

The distribution of the predicted annual mean DAOD (Fig. 5 (b)) agrees well with the IASI result (Fig. 5 (a)). The most notable discrepancy is an overestimation by the predictions over the Sahara. Here, the mean values of the IASI retrievals are significantly affected by small or even negative values between the major dust outbreaks, where the model usually predicts
small but non-zero values which accumulate to a larger annual average.

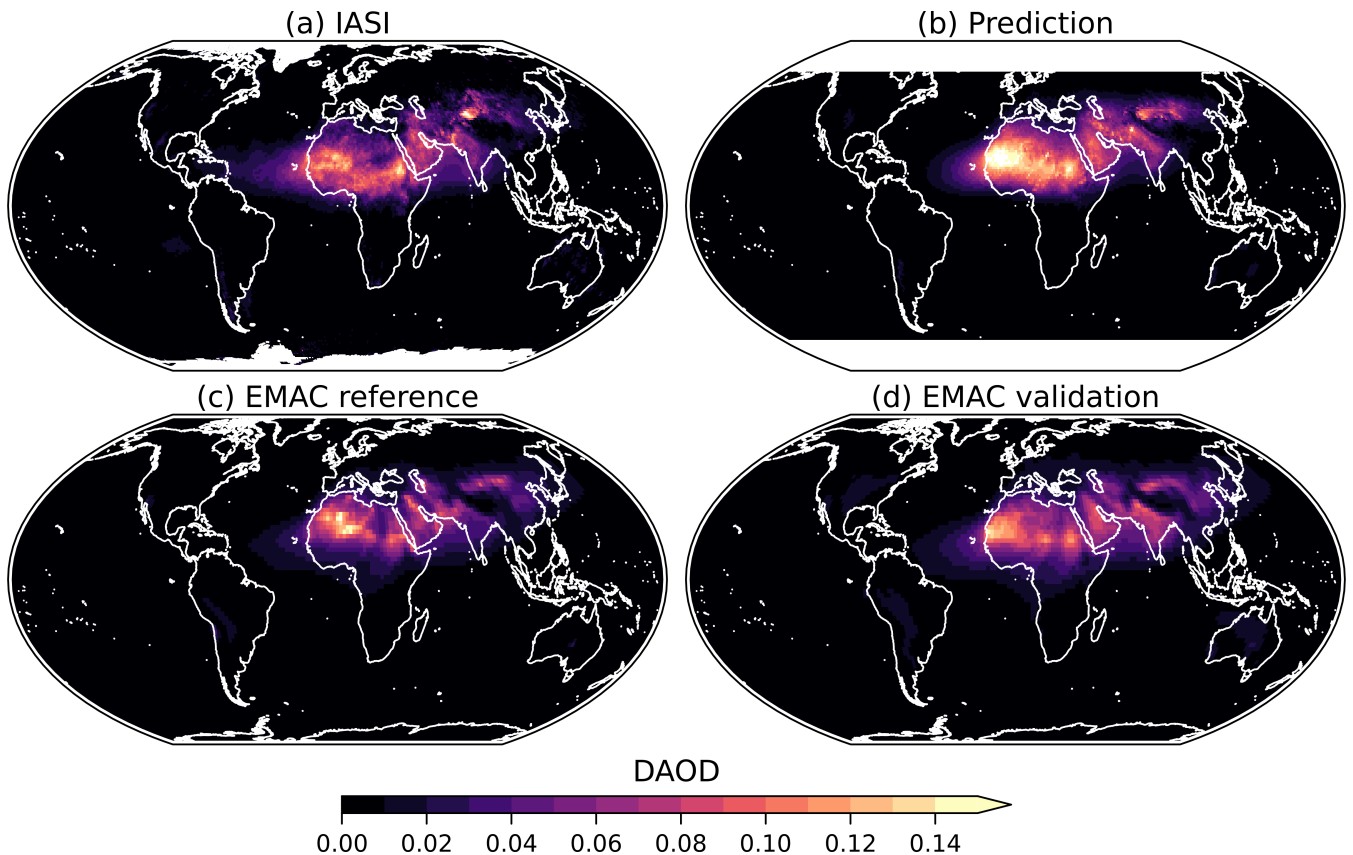

**Figure 5.** Annual mean 10 $\mu$m DAOD during the year 2009 observed by IASI (a), predicted by the trained dust model (b), and simulated by EMAC with reference (c) and trained (d) mineral emissions.

We conclude that the model generally reproduces the main features of the global dust distribution. This makes it suitable for the application in climate models, as discussed in the next sections. Additionally, valuable information can be obtained directly from analysing the model.

To study the relative contribution of different source regions to the global dust emissions, we compute hourly dust emission
fluxes by evaluating the dust emission submodel (light grey box in Fig. 1) from the trained model stand-alone. Accumulating the emission fluxes from the 9 major dust source regions considered by Kok et al. (2021) over the test period (year 2009) yields the fractional contributions to the global dust emissions shown in Table 1. These contributions are well within the range of the models considered by Kok et al. (2021), with a comparably large contribution from the Middle East and Central Asia.

To estimate the importance of the individual input variables, we made use of the skill score $S$ defined by Taylor (2001),

$$S = \frac{4(1+r)^4}{(\sigma_1/\sigma_2 + \sigma_2/\sigma_1)^2(1+r_0)^4},$$ (6)

**Table 1.** Fractional contribution of major source regions to the global mineral dust emissions.

| Region | Contribution to global emissions | Coordinates |
|---|---|---|
| Western North Africa | 21 % | 20°W to 7.5°E, 18 to 37.5°N |
| Eastern North Africa | 18 % | 7.5 to 35°E, 18 to 37.5°N |
| Southern Sahara and Sahel | 13 % | 20°W to 35°E, 0 to 18°N |
| Middle East/Central Asia | 26 % | 35 to 75°E, 0 to 35°N and 35 to 70°E, 35 to 50°N |
| East Asia | 8.4 % | 70 to 120°E, 35 to 50°N |
| North America | 1.7 % | 130 to 80°W, 20 to 45°N |
| Australia | 3.2 % | 110 to 160°E, 10 to 40°S |
| South America | 1.9 % | 80 to 20°W, 0 to 60°S |
| Southern Africa | 1.1 % | 0 to 40°E, 0 to 40°S |

where $r$ is the correlation coefficient and $\sigma_1$ and $\sigma_2$ are the standard deviations of modelled and observed values. As maximum attainable correlation coefficient we simply used $r_0 = 1$ since we focus on relative changes of the skill score (a more accurate estimate $r_0 < 1$ would result in higher skill scores).

We computed the DAOD during the test period multiple times, each time replacing the values of one input variable with random values sampled from the probability distribution of the same variable, or, in the case of topography, using its mean value (695 m) as a constant input (random values would correspond to an unrealistic terrain). Subtracting the resulting skill scores from those obtained using the full input data yields the impact $\Delta S$ of each variable, which indicates the importance of the variable to the model predictions. Figure 6 identifies soil moisture and surface friction velocity as important input parameters. Due to the slightly different methodology, the exact value obtained for the topography should not be directly compared with the other values, but the use of topographic data clearly improves the model predictions. In contrast, the effect of snow is small because conditions where snow suppresses emissions that are not already suppressed by other input variables are rare. We assume that the importance of the variables for the model predictions is to some extent representative of their real importance for dust emissions. This assumption neglects the interdependence of the input variables as well as the effect of model approximations and needs to be verified by detailed analysis in future studies. Nevertheless, the importance of soil moisture in our model is remarkable and consistent with the conclusions of Klingmüller et al. (2016).

Similarly, the dependence of modelled emissions on input parameters should indicate realistic relationships. As an example, the dependence of emissions on the two local input variables with the greatest influence on the skill score, surface friction velocity and soil moisture, was assessed (Fig. 7). Under ideal conditions, i.e. without soil moisture, vegetation and snow, emissions increase only slowly with surface friction velocity until they increase more rapidly from about $0.2\,\mathrm{ms^{-1}}$. As soil moisture increases, emissions decrease until they vanish at a volumetric soil moisture of about 0.3. Like the parameter importance, this analysis will be addressed in detail in a separate study.

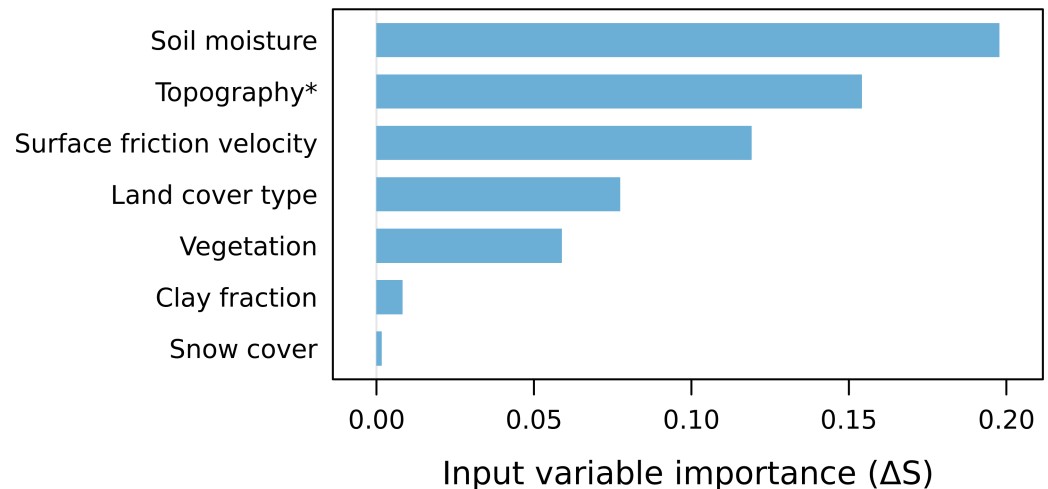

**Figure 6.** The importance of the individual input variables, estimated by their impact on the model skill score $S$. For each variable, the impact was obtained by comparison with a model evaluation replacing the variable with random or, in case of the topography, constant values.

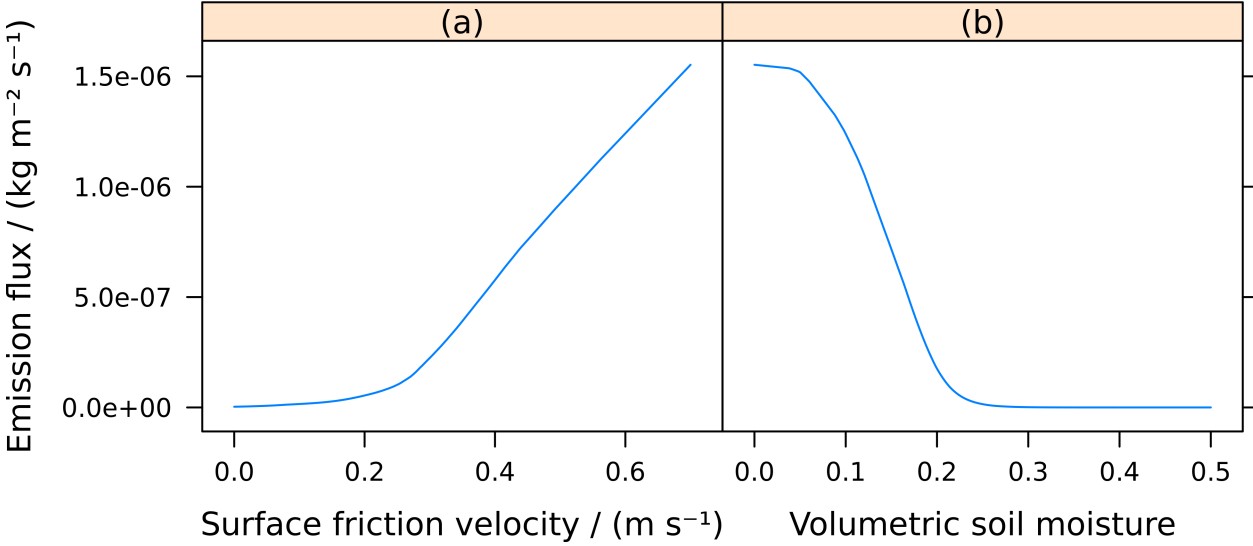

**Figure 7.** The modelled emission flux as a function of surface friction velocity (a, without soil moisture) and soil moisture (b, with surface friction velocity 0.7 m $s^{-1}$). The flux was computed assuming the absence of vegetation and snow and an average clay fraction (20.6 %). The topography and land cover factors were set to unity and the normalised described in Section 6 was applied.

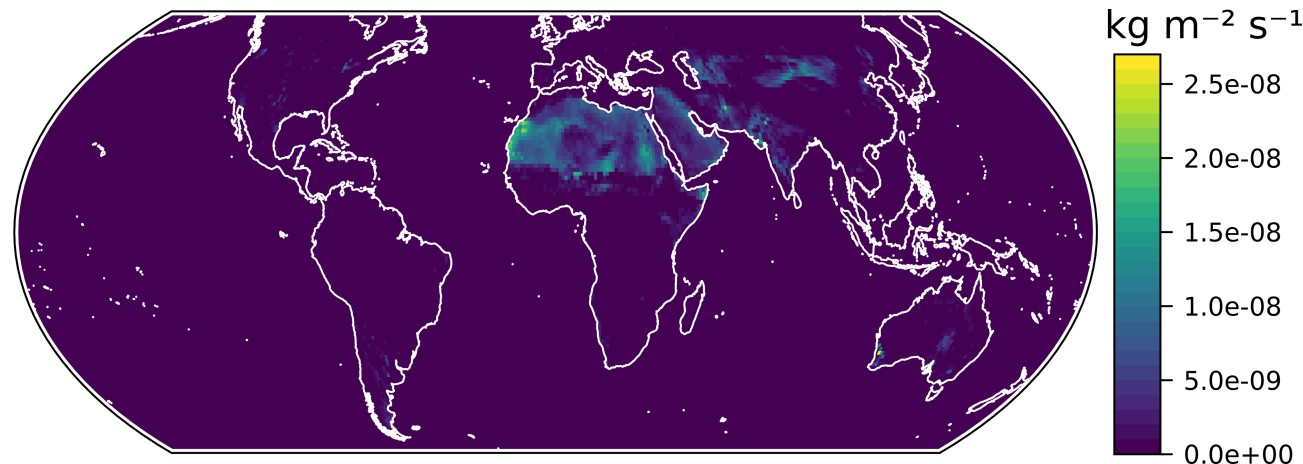

**Figure 8.** Mean aeolian dust emission flux during the test period.

## 6 EMAC setup

The output of the emission submodel corresponds to the change of the DAOD (weighted by the grid cell area and the standard deviation of the DAOD). Avoiding assumptions on the size distribution and optical properties of the mineral dust particles and consistent with our simplifying assumption that dust extinction and concentration are proportional, we converted the hourly emission flux output to a mass concentration flux in kg m$^{-2}$ s$^{-1}$ using a single tuning parameter chosen such that the global annual emission for 2008 amounts to 4.3 Gt which is the upper limit of the range reported by Huneeus et al. (2011). This choice represents a relatively high global total emission. However, since our model does not use any a priori emission mask, also regions with locally very low emissions contribute to the global total emissions, which significantly enhances the value due to the large area of those regions. Our choice for the total emission value is supported by the realistic DAOD levels we obtain over the dust belt. The mean of the resulting emission flux during the test period 2009 is shown in Fig. 8.

The resulting dust emission flux data were used in a validation simulation with the ECHAM/MESSy atmospheric chemistry–climate model (EMAC) (Jöckel et al., 2006). We employed a recent EMAC version based on release 2.55.2. A list of the activated EMAC submodels including references is provided in Table A1 in appendix A.

The total dust emission flux was distributed over the accumulation and coarse modes consistently with the online dust emission scheme, where 94.7 % of the dust is emitted into the coarse mode. The modes are represented by log-normal distributions with fixed geometric standard deviations ($\sigma_\mathrm{g} = 1.59$ for the accumulation and $\sigma_\mathrm{g} = 2$ for the coarse mode). The count median diameter of each mode can vary with a fixed threshold between accumulation and course mode at 1.4 $\mu$m. Accordingly, the percentage of fine particles in the emissions is comparable to the 4.3 % (95 % conficence interval 3.5 to 5.7 %) found by Kok et al. (2017) for particles smaller than 2 $\mu$m. There is no strict upper limit for the size of coarse particles, but super coarse particles with diameters exceeding 20 $\mu$m are only represented by the upper tail of the log-normal coarse mode.

The simulation covered three years from 2008 to 2010, where the year 2008 is used as model spin-up period. The results for the years 2009 and 2010 are used to evaluate the performance of the trained dust emissions during test and training periods, respectively.

Horizontally, the setup used a Gaussian T63 grid with approximately 1.9° latitude/longitude spacing. The simulation assimilated observational data by nudging temperature, vorticity and divergence above the boundary layer to meteorological reanalysis data of the ECMWF and by using the sea surface temperature from the same dataset.

For comparison, we performed a reference simulation with the same settings but using the online dust emission scheme of Klingmüller et al. (2018) instead of the trained emissions.

# 7 EMAC evaluation

The aerosol optical depth (AOD) at 10 $\mu$m wavelength is dominated by coarse aerosol particles which in turn are dominated by mineral dust and sea salt particles. Accordingly, we obtain the EMAC DAOD by weighting the 10 $\mu$m AOD with the volume fraction of dust in the local dust (DU, 2.65 g cm$^{-3}$) and sea salt (SS, 2.17 g cm$^{-3}$) mixture, $\text{DAOD}(10\mu\text{m}) = \text{AOD}(10\mu\text{m})\, V_{\text{DU}}/(V_{\text{DU}} + V_{\text{SS}})$.

The results for the annual mean 10 $\mu$m DAOD for the year 2009 from the reference and validation simulations are shown in Fig. 5 (c) and (d). Both simulations yield a DAOD distribution similar to that obtained from IASI. Note that the model results are obtained from daily average values and thus consider data for all model time steps of the year, whereas the IASI result is based on the instantaneous values which are retrieved at most twice a day and include a substantial amount of missing values, therefore the IASI distribution is more noisy and has a larger year-on-year variability.

The fact that the trained emissions reproduce the high DAOD values that are expected over the dust belt, but low levels elsewhere, is quite remarkable as the model architecture treats all regions worldwide equally and there is no a priori selection of dust source regions involved. Moreover, the spatial pattern of the DAOD resulting from the trained emissions agrees more closely with the IASI observation than the reference result. The correlation coefficient of the modelled annual grid cell DAOD values and the corresponding IASI values (re-sampled to the model grid) is 0.82 for the reference simulation, but 0.88 for the trained emissions.

The annual mean AOD at 550 nm from the reference and validation simulations are compared to the corresponding MODIS observations in Fig. 9. Again the result based on the trained emissions is closer to the observations than the reference results. The correlation coefficient of the modelled annual grid cell AOD values and the corresponding MODIS values (re-sampled to the model grid) is 0.72 for the reference simulation, and 0.73 for the trained emissions.

To complement the satellite observations, we considered ground based observations by Aerosol Robotic Network (AERONET) stations in the 7 dust affected regions defined by Klingmüller et al. (2018) (Fig. 10). We quantified the degree of agreement of AERONET and EMAC daily AOD values using the Pearson correlation coefficient and the skill score $S$. The AERONET data was interpolated to the 550 nm wavelength of the EMAC output using the Ångström exponent.

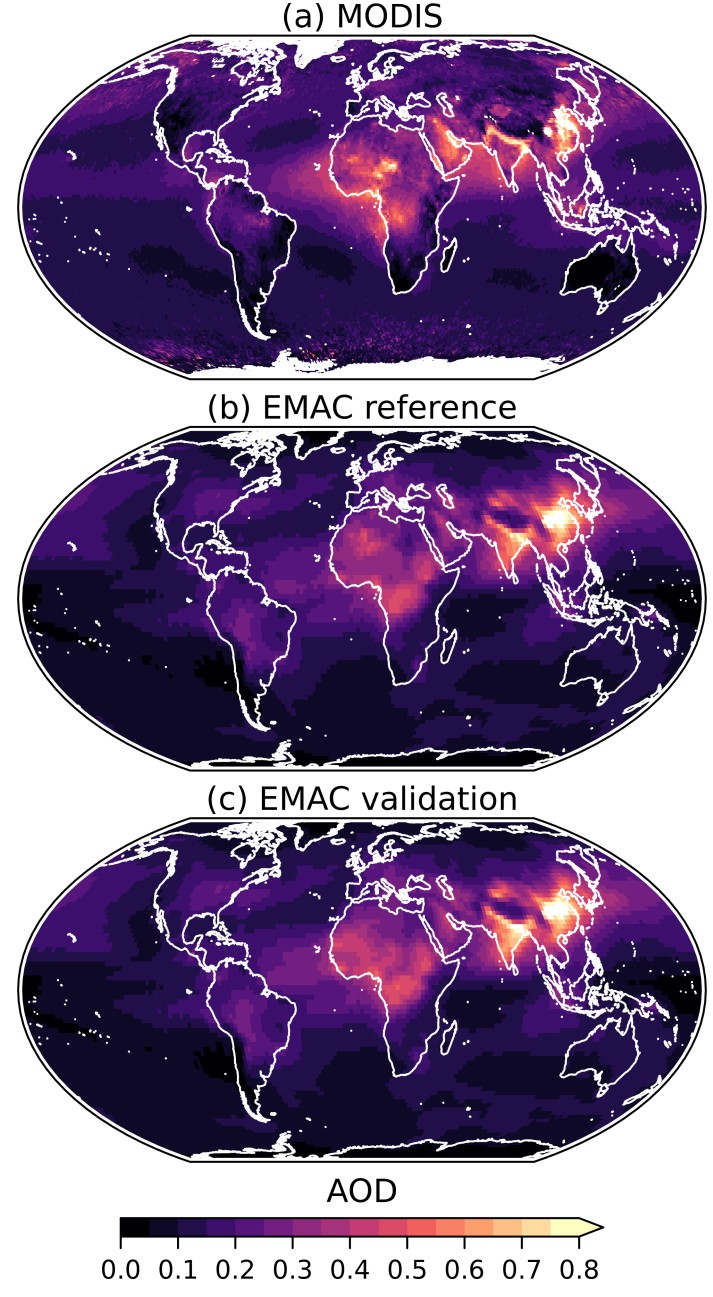

**Figure 9.** Annual mean 550 nm AOD during the year 2009 observed by MODIS (a) and simulated by EMAC with reference (b) and trained (c) mineral dust emissions.

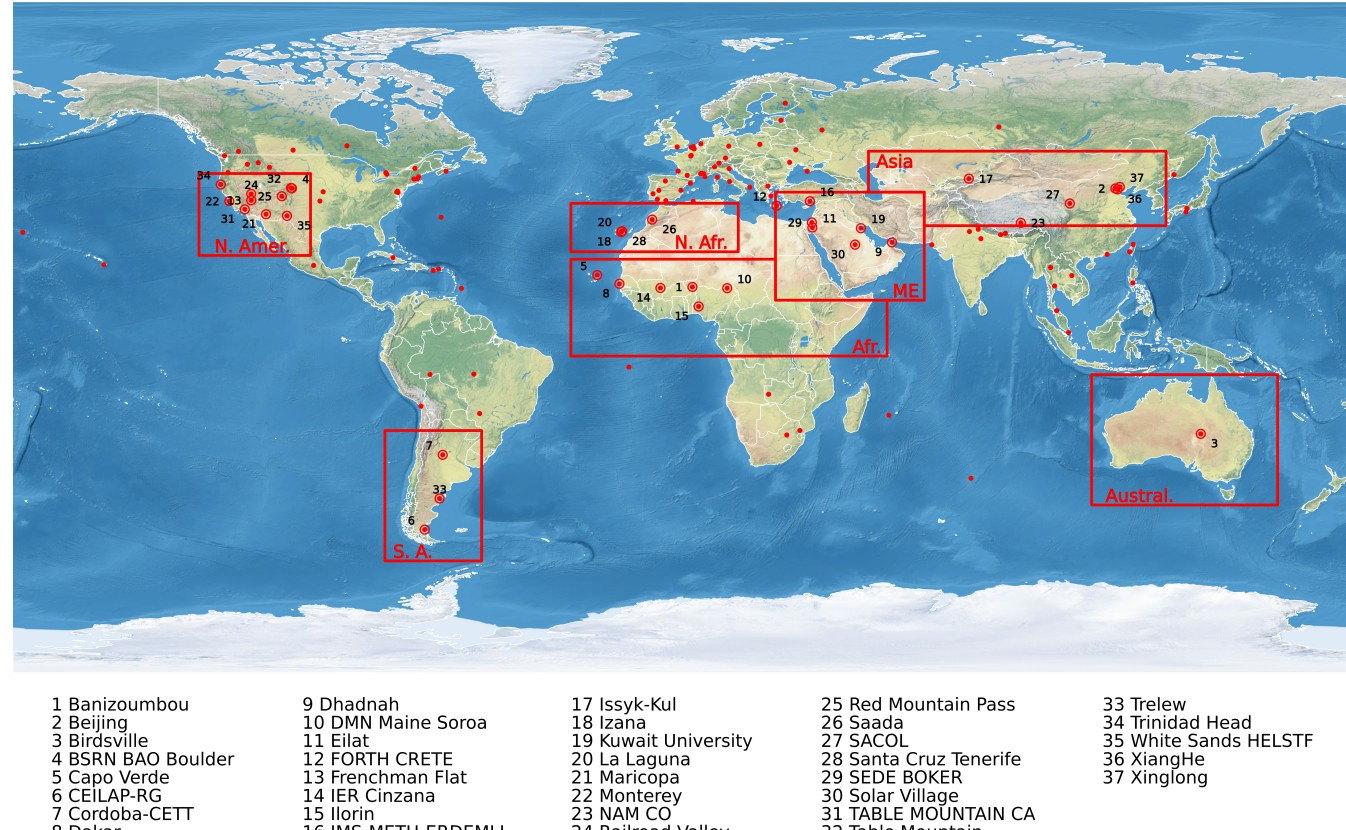

| | | | | |
|---|---|---|---|---|
| 1 Banizoumbou | 9 Dhadnah | 17 Issyk-Kul | 25 Red Mountain Pass | 33 Trelew |
| 2 Beijing | 10 DMN Maine Soroa | 18 Izana | 26 Saada | 34 Trinidad Head |
| 3 Birdsville | 11 Eilat | 19 Kuwait University | 27 SACOL | 35 White Sands HELSTF |
| 4 BSRN BAO Boulder | 12 FORTH CRETE | 20 La Laguna | 28 Santa Cruz Tenerife | 36 XiangHe |
| 5 Capo Verde | 13 Frenchman Flat | 21 Maricopa | 29 SEDE BOKER | 37 Xinglong |
| 6 CEILAP-RG | 14 IER Cinzana | 22 Monterey | 30 Solar Village | |
| 7 Cordoba-CETT | 15 Ilorin | 23 NAM CO | 31 TABLE MOUNTAIN CA | |
| 8 Dakar | 16 IMS-METU-ERDEMLI | 24 Railroad Valley | 32 Table Mountain | |

**Figure 10.** Location of the AERONET stations used for the model evaluation.

Figure 11 reveals that the validation simulation using the trained emissions achieves substantially higher skill scores and correlation coefficients over Africa and the Middle East than the reference simulation. Over regions where the atmospheric aerosol is less dominated by aeolian dust, including Asia, the Americas and Australia, the effect of the new emissions is smaller but generally enhancing both, the skill scores and the correlation coefficients.

Figure 12 compares the modelled annual mineral dust deposition to observations at different sites worldwide (Huneeus et al., 2011; Checa-Garcia et al., 2021). Both, the correlation coefficient $r$ and the skill score $S$ are higher for the validation simulation, whereas the root-mean-square error (RMSE) and the mean average error (MEA) are smaller. All metrics indicate that the data-driven emission scheme used in the validation simulation results in a more realistic global distribution of the mineral dust deposition than in the reference simulation.

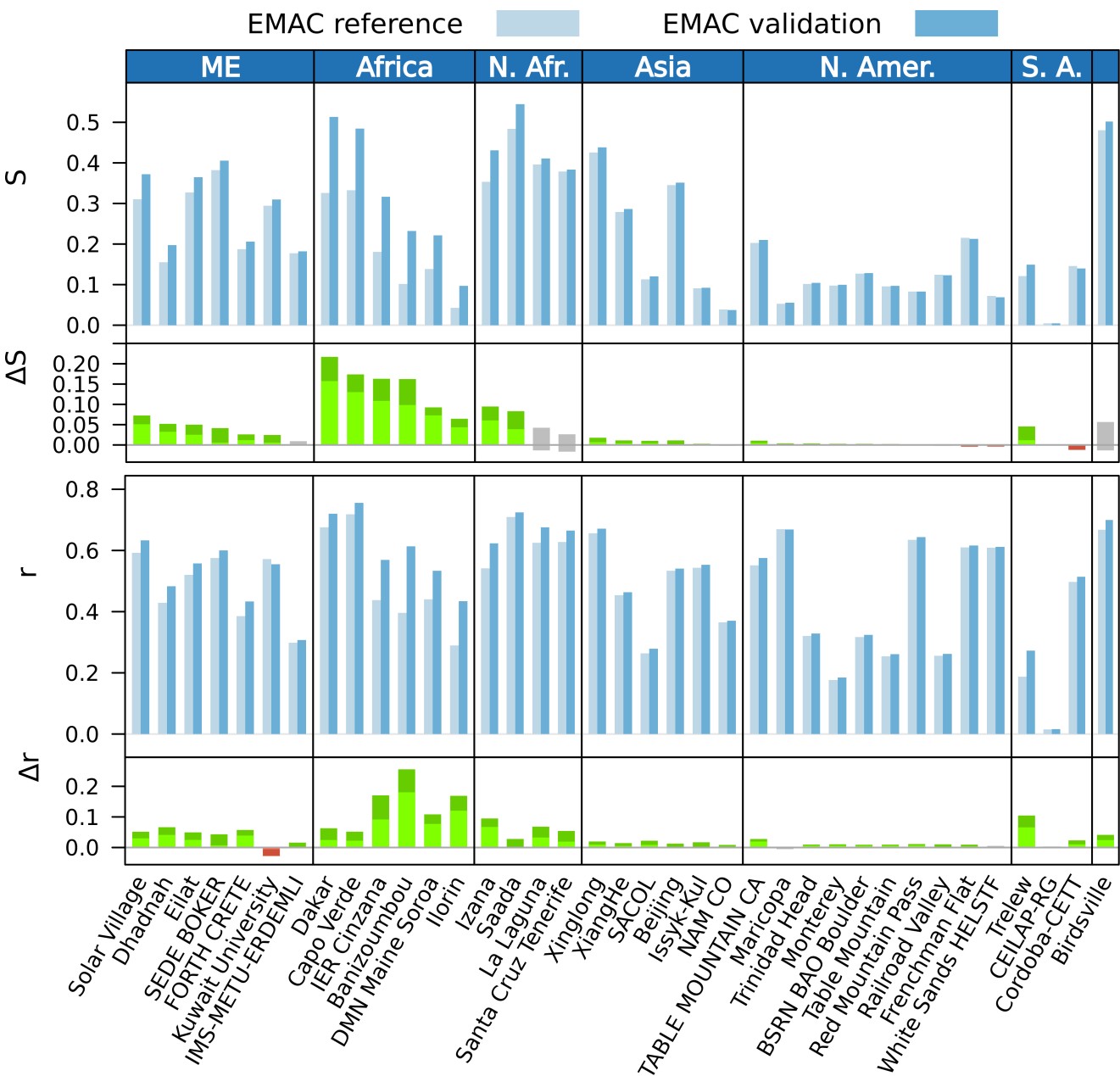

**Figure 11.** Skill score $S$ and correlation coefficient $r$ comparing the daily AOD simulated by EMAC with AERONET observations. The AERONET data was interpolated to the 550 nm wavelength of the model output using the Ångström exponent. The red, green and grey bars depict the differences between the results for the trained emissions and for the reference emissions, with green bars indicating that the results for the trained emissions agree more closely with the measurements by at least 1 standard deviation ($\sigma$). The corresponding error intervals are indicated by darker colours.

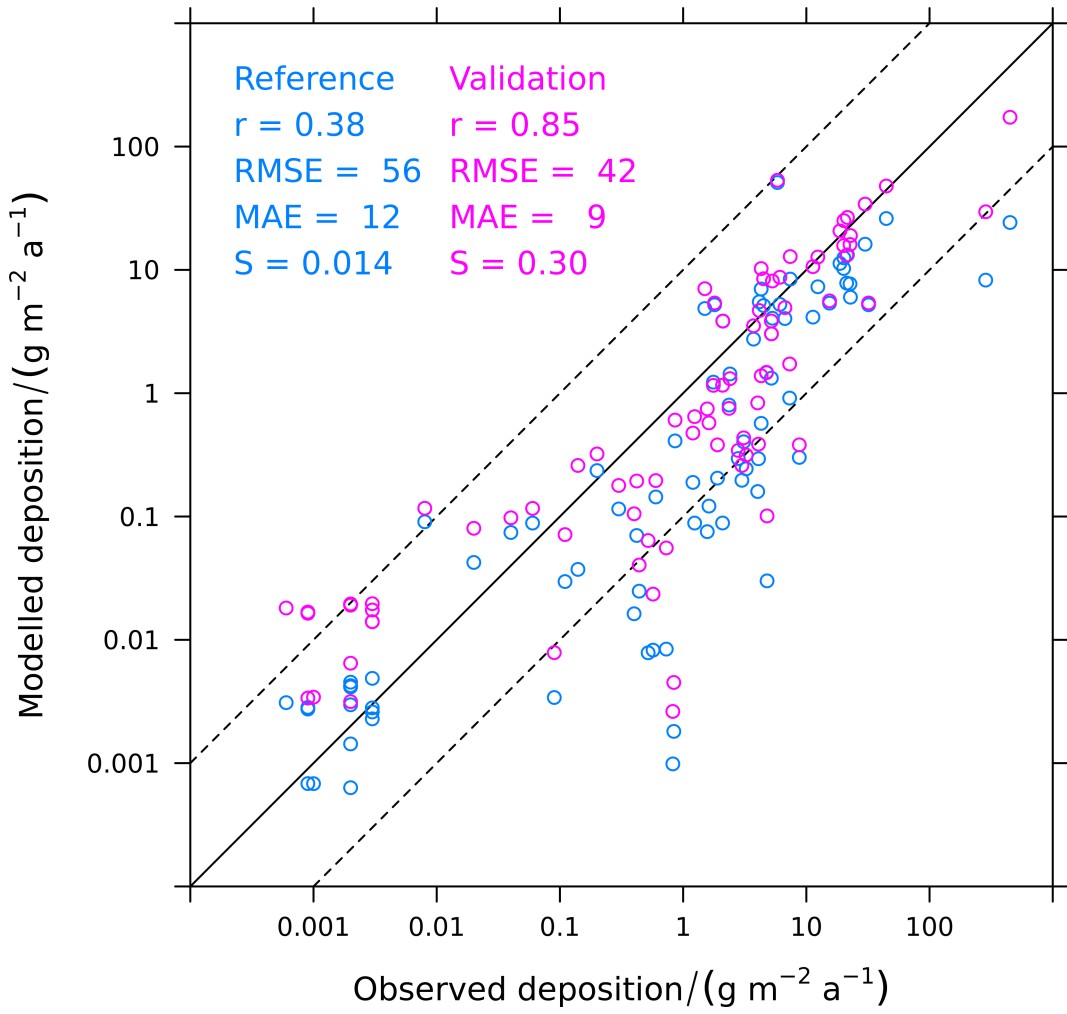

**Figure 12.** Comparison of the mineral dust deposition simulated by EMAC and the observations from the AeroCom dust benchmark data set (Huneeus et al., 2011). Correlation coefficient $r$, root-mean-square error RMSE, mean absolute error MAE and skill score $S$ are provided for both, the reference simulation and the validation simulation using the trained dust emissions.

# 8 Conclusions

We presented a trainable aeolian dust model which successfully learned from satellite observations to compute realistic global dust distributions and emissions based on the environmental conditions. On the one hand, the model architecture incorporates little a priori assumptions about the nature of dust sources, on the other hand we explicitly modelled the better known components of the dust cycle, in particular the atmospheric transport.

We focussed on the emission submodel of the trained model, representing a novel dust emission scheme which was derived independently of existing methods. Using the EMAC model as example, we demonstrated its superior performance when applied within an atmospheric chemistry-climate model.

Future developments will include model improvements by adjusting hyperparameters and resolution, addressing characteristics of the DAOD data set through preprocessing and model adjustments, considering additional input data (e.g., on biological soil crusts, Rodriguez Caballero et al., 2022), and utilising related machine learning models (e.g., for soil moisture, Klingmüller and Lelieveld, 2021).

The analysis of data-driven models will provide new insights into the nature of dust mobilisation processes and their dependency on environmental conditions. This might also allow to refine existing parametrisations of mineral dust emissions, e.g., the surface friction velocity or soil moisture terms.

Relating emission sources or surface concentrations to observed column burdens of gases and particles, being influenced by transport and other processes at the surface and in the atmosphere, is an emerging challenge in applications of satellite observations. The concept presented here in the context of aeolian dust can be generalised and applied to other atmospheric constituents for which satellite retrievals are available, for example additional aerosol components, nitrogen oxides or water vapour.

## Appendix A

The submodels used in the EMAC simulations are listed in Table A1.

*Code and data availability.* The data-driven dust model and all relevant data are available in the Edmond Open Research Data Repository of the Max Planck Society (https://doi.org/10.17617/3.XPDIES). The ECHAM climate model is available to the scientific community under the MPI-M Software License Agreement (https://mpimet.mpg.de/en/science/modeling-with-icon/code-availability, last access: 8 September 2022, MPI-M, 2022). The Modular Earth Submodel System (MESSy) is continuously further developed and applied by a consortium of institutions. The usage of MESSy and access to the source code are licensed to all affiliates of institutions which are members of the MESSy Consortium. Institutions can become a member of the MESSy Consortium by signing the MESSy Memorandum of Understanding. More information can be found on the MESSy Consortium Website (https://www.messy-interface.org, last access: 8 September 2022, MESSy, 2022). Commit 869808b6 in the MESSy source code repository was used, which is based on MESSy version 2.55.2. All relevant features are included in the "devel" branch and will be available in the next official release.

**Table A1.** EMAC submodels used in the present study

| Submodel | Domain | Reference |
| --- | --- | --- |
| AEROPT | Aerosol optical properties | Lauer et al. (2007); Klingmüller et al. (2014) |
| AIRSEA | Air-sea exchange | Pozzer et al. (2006) |
| BIOBURN | Biomass burning emissions | Kaiser et al. (2012) |
| CLOUD | Cloud physics | Jöckel et al. (2006) |
| CLOUDOPT | Cloud optical properties | Dietmüller et al. (2016) |
| CONVECT | Convection | Jöckel et al. (2006) |
| CVTRANS | Convective transport | Jöckel et al. (2006) |
| DDEP | Dry deposition | Kerkweg et al. (2006a) |
| E5VDIFF | Vertical diffusion | |
| GMXe | Aerosol microphysics | Pringle et al. (2010) |
| GWAVE | Gravity waves | |
| JVAL | Photolysis | Sander et al. (2014) |
| LNOX | Lightning NO$_x$ | Tost et al. (2007) |
| MECCA | Gas-phase chemistry | Sander et al. (2019) |
| OFFEMIS | Offline emissions | Kerkweg et al. (2006b) |
| ONEMIS | Online emissions | Kerkweg et al. (2006b) |
| ORBIT | Orbital parameters | Dietmüller et al. (2016) |
| ORACLE | Organic aerosol composition and evolution | Tsimpidi et al. (2014) |
| RAD | Radiative transfer | Dietmüller et al. (2016) |
| SCAV | Scavenging | Tost et al. (2006) |
| SEDI | Sedimentation | Kerkweg et al. (2006a) |
| SURFACE | Surface parametrisations | |
| TNUDGE | Tracer nudging | Kerkweg et al. (2006b) |

*Author contributions.* KK conceived and conducted the study and wrote the manuscript with contributions from JL. Both authors discussed the results and finalised the article.

*Competing interests.* KK is member of the editorial board of Geoscientific Model Development. The peer-review process was guided by an independent editor, and the authors have also no other competing interests to declare.

*Acknowledgements.* KK acknowledges support from the Max Planck Graduate Center in Mainz and the MaxWater Initiative of the Max Planck Society.

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
