# Peer review of "Data-driven aeolian dust emission scheme for climate modelling, evaluated with EMAC 2.55.2"

_Geoscientific Model Development, 2022_

## Author Comment (AC1)

**Reply to RC1**

We thank the referee for the very helpful comments. We have revised the manuscript accordingly. In the following please find our replies to the individual comments.

> This manuscript from Klingmüller and Lelieveld presents a dust scheme that leverages techniques from machine learning (ML) to help represent dust emitted from arid and semiarid regions. The model setup is well described and the comparison with observations is sufficiently clear to appreciate regions where ML improves upon the classical representation of dust emission that is currently used in the EMAC model used by this group. The paper is sufficiently well written to warrant publication but I would like to see minor questions answered that can improve on it and help the reader follow the choices that have been made in this work.
> A first thing that should be explained since it is can be seen as both a wise or a curious choice is why the authors choose to compare the model results to the dust aerosol optical depth at 10 mm instead of the more classical comparison at a wavelength of 550nm.

We have added the motivation "In the infrared, coarse particles scatter radiation more efficiently than submicron particles. Therefore, observations at a wavelength of 10 $\mu$m are most sensitive to mineral dust and sea salt and are little affected by other aerosols, resulting in a particularly reliable DAOD retrieval."

> Lines 73-75: please explain to the reader why you need to normalize surface friction velocity, soil moisture, snow depth, KAI, geopotential and clay fraction, is it inherent to the way the ML technique is used? It is hard for the reader to guess.

We have added the explanation "This normalisation maps all input variables and associated gradients into a similar and relatively small range to support the model parameter optimisation routine during the training process."

> Line 109: you imply that the dust emissions are injected in the first 4 lowest layers of the model, if that is the case, please give the rational of why you chose to do that instead of injecting dust in the lowest model layer and what are the averaged heights of these first 4 layers.

The dust emissions are only injected into the lowest of the 4 layers, which is now more clearly stated.

> Line 123: the text seems to indicate that aerosol wet scavenging is a function of total precipitation in a model gridbox. Physically, this is not the choice since aerosols are scavenged as a function of the amount of precipitation formed in the aerosol layer and is also dependent on the rate of precipitation coming from above. Please explain better the choice made here.

Considering only the leading order effect that more precipitation means more scavenging is one of the simplifications to reduce the complexity of the model. The motivation is that, due to the transport between emissions and scavenging, primarily the main statistical properties of

the latter influence the emissions submodel during training. In retrospect, this seems justified by the good performance of the trained emissions in the EMAC model with its very detailed wet scavenging parametrisation. We have added: "This approximation considers the correlation between precipitation and dust removal as the dominant statistical relationship with the strongest influence on dust source parameters during training. More detailed parametrisations could improve future versions of the model".

> Line 143: this is the first time you mention f_loss which is defined below in line 155. You should at least introduce what f_loss represents before this line.

We now avoid mentioning $f_{\text{loss}}$ in this line.

> Lines 177-178: "The temporal correlation coefficients of the observed and predicted hourly DAOD values within each grid cell are typically greater than 0.5 over the regions affected by desert dust (Fig. 3)." You give the impression to the reader that the correlation coefficients are always above 0.5, as you describe later on in the paper it might be the case for spring and summer and it is not the case for the 2 other seasons.

The temporal correlation discussed in Lines 177f considers all seasons and is typically greater than 0.5 in the dusty regions (Fig. 3). Later on, in lines 183ff, we discuss the spatial correlation, which is typically greater than 0.5 during the dusty seasons. We have added inline equations to make this more clear.

> Caption of Figure 1: Please spell out that MAE stands for mean absolute error and RMSE stands for random means square error.

We now define all 4 numbers shown in the plot in the caption.

> Color chart in Figure 3: the use of red and saturated red make it difficult to appreciate the differences between regions that have a correlation coefficient of 0.5 compare to 0.7 or even 0.9. Please take a color scale that allows to appreciate this differences more accurately.

We have increased the contrast.

> Line 198: To appreciate an annual mean emission of 4.3 Gt/yr it would be informative to give the fraction of this emission that are particles below a diameter of 1mm since they will influence much more the shortwave and Kok et al., (2017) have established a constraint on this fraction as well as on the total emission. Please indicate what is the cutoff of the dust size distribution in your model for comparison with other models. It would be of interest to know how much you emit for the larges regions emitting dust (see paper by Kok et al 2021)

We have added details about the accumulation and coarse mode parameters which clarify that the accumulation mode (which receives 5.3 % of the dust emissions) has a count median diameter $< 1.4 \ \mu$m and that there is no strict upper limit for the particle size in the coarse mode (but no

additional mode for super coarse particles). We now also present the fractional emissions from the regions considered by Kok et al. 2021 in the new Table 1.

> Line 219: When you explain how EMAC DAOD is obtained when dust and seasalt are present, you should indicate the assumptions made for the density of dust and of seasalt to allow other researchers to make a comparable evaluation.

We have added the densities.

> Color bar of Figure 7: you should extend this color bar as the AOD scales of 0.0, 0.1,... 0.8 are too close one another to be legible.

We have extended the colour bar.

> You could have pushed further the comparison with observations by comparing yearly mean dust deposition over the globe, this is done et Checa-Garcia et al., (2021) for instance.

We have added a comparison with deposition observations in the new Fig. 12.

> Thank you for this interesting contribution.
> References:
> Kok et al., 2017, Smaller desert dust cooling effect estimated from analysis of dust size and abundance, Nat com., doi: 10.1038/ngeo2912
> Kok, J. F., Adebiyi, A. A., Albani, et al., 2021: Contribution of the world's main dust source regions to the global cycle of desert dust, Atmos. Chem. Phys., 21, 8169–8193, https://doi.org/10.5194/acp-21-8169-2021.
> Checa-Garcia, R., Balkanski, Y., Albani, S., et al.: Evaluation of natural aerosols in CRESCENDO Earth system models (ESMs): mineral dust, Atmos. Chem. Phys., 21, 10295–10335, https://doi.org/10.5194/acp-21-10295-2021, 2021.

---

## Author Comment (AC2)

**Reply to RC2**

We thank the referee for the constructive review. We have addressed the comments with a revised manuscript and the point-by-point reply below.

> Review of "Data-driven aeolian dust emission scheme for climate modelling, evaluated with EMAC 2.54" By Klingmüller and Lelieveld
>
> The manuscript presents a data-driven (machine learning) dust emission scheme that is trained together with other data-driven components (transport, deposition,...) to reproduce dust aerosol optical depth retrievals by the Infrared Atmospheric Sounding Interferometer on board the MetOp-A satellite. The resulting data-driven scheme is then applied in a climate chemisty model (EMAC) and compared to the standard dust emission scheme in that model. The results show improvement in the representation of the dust cycle when using the data-driven scheme in comparison to AERONET measurements. This is a very interesting and novel contribution that is well-written and deserves to be published. I have some comments that would need to be addressed before the paper is accepted for publication. (I have to say that I am an expert on dust emission and modeling and I have limited knowledge on machine learning. Therefore, the paper would benefit from an additional review by a machine learning expert.)
>
> ■ The application of data-driven (learning) methods to dust emission and modeling is quite unprecedented. There is almost no literature on the subject and generally dust modelers (like me) are not specialized in machine/deep learning methods. I find that the description of the method lacks detail, both general/conceptual and specific, at least the kind of details that would benefit the future main readership of this article (dust and climate modelers like me). It would therefore be very valuable that the authors add more detail to the background and the methodology. Why they selected such learning framework and not others? Is this a deep learning algorithm, neural network? What does it bring to the table compared to other machine methods for this specific problem? What functionalities of Pytroch were used? This is really not totally clear and it is not explicitly discussed.

We have expanded Section 3 to better explain and motivate the methodology.

> ■ The derived emission scheme depends upon the following inputs: friction velocity, vegetation, clay fraction, soil moisture, snow cover, topography and land use cover. All of them depend upon location and some of them on location and time. However, if I correctly understood, with this method, emission will depend upon grid-cell (location) even if all the inputs are the same. Can you elaborate on that? I am assuming that this is a feature of the (deep) learning method used, but maybe just I misunderstood the method (sorry if that is the case). This question goes back to my main point: more details are needed to understand the data-driven model and its implications. This issue is potentially important: I wonder about the determinism of the method. Do you obtain different emissions in different grid cells even if the input parameters are exactly the same? Particularly, in climate model simulations (future and past climates), with changes in climate regime, the lack of determinism could be problematic. If dust emission is not deterministic, could you provide some data analysis to show to what extent this could be

(or not) problematic. In other words, could you show the degree of determinism of your model? One way would be to run one time step of the model with constant inputs everywhere.

The emission model depends on the local values of the above mentioned input variables and does not explicitly depend on the location and time, which we have clarified in the revised Section 3 (only the topography factor additionally considers the terrain in surrounding grid cells). Consequently, if evaluated at a different location and time but with the same input values, it yields identical emissions. This makes the emission scheme well suited for climate scenario simulations because it does not rely on memorised spatial or temporal patterns that might change with climate.

■ What is the relative importance of the different features/inputs used? Can that be derived from your method? All in all, as you see I am really interested on this work but there is a general lack of diagnostics to understand the behavior of the system.

In the revised manuscript, we have included estimates of the importance of each input variable (Fig. 6). These estimates should be interpreted with caution, in particular because the input variables are to some extent correlated, so that the effect of one variable may be reflected in the importance estimate of another variable. A more detailed analysis of the importance of different factors is subject of a follow-up study.

■ There are many choices that seem a bit arbitrary or at least not justified. Some examples follow. Why the atmosphere goes up to 4 km and there are only 4 vertical layers? Why the dust concentration in the four layers is increased by the emission flux? The dusty PBL in the Sahara and the Saharan air layer reach easily 6 km in Summer.

In section 2 we now explain the choice of the vertical levels: "Mineral dust can reach higher altitudes, but since most of the dust mass remains within these layers, this approximation is a reasonable compromise between a realistic representation of dust transport and an acceptable computational burden." Future versions, especially when utilising new hardware, will consider more and higher levels. The dust emission flux increases the concentration in the lowest layers directly, the concentration in the higher levels increases only by vertical transport. We have clarified the sentence of Eq. (3).

■ Line 218: The model seems think that dry deposition is unimportant. Can you comment on that? There is strong dry deposition over sources. Is it possible that the method is just adjusting/providing a net emission (emission – deposition)? That could be the case as dry deposition scales with friction velocity (like emission). Comment on the implications.

It is plausible that the emission flux we obtain is to some extend a net emission flux because deposition contributions that are directly linked to the conditions causing the emissions, i.e., that have the same spatial and temporal distribution, cannot be distinguished from a reduction of the emissions. However, since the distribution of deposition in general is not identical to that of the emissions, the model should be able to learn deposition and emission parameters

independently. We have rephrased line 121 to "[...] indicating that the current implementation of our approach is not sensitive enough to identify dry deposition. The reason is likely the limited atmospheric residence time discussed below, which represents an unrealistic additional removal process. Moreover, a fraction of the deposition that always occurs collocated with emissions might be represented by a reduced emission flux."

> ■ Have you analyzed the trainable parameters? It would be very informative to understand whether the trainable parameters make physical sense. In my previous comment, clearly the fact that there is no dry deposition outlines this problem.

Besides creating an emission scheme for the application in climate models, obtaining a better understanding of dust mobilisation by studying the trained model is a main goal of our efforts. However, the latter requires careful analysis. One reason is that, as mentioned above, our present input variables are not independent, which is not a problem with regard to good model predictions, but complicates studying the effect of the individual variables. Therefore, in this article we focus on the practical application of the model and address a comprehensive model analysis in a separate article. Nevertheless, we have added plots of the effect of the surface friction velocity and the soil moisture, which, according to our preliminary analysis, are particularly important.